# A C-Arm-Free Minimally Invasive Technique for Spinal Surgery: Cervical and Thoracic Spine

**DOI:** 10.3390/medicina59101779

**Published:** 2023-10-06

**Authors:** Masato Tanaka, Konstantinos Zygogiannnis, Naveen Sake, Shinya Arataki, Yoshihiro Fujiwara, Takuya Taoka, Thiago Henrique de Moraes Modesto, Ioannis Chatzikomninos

**Affiliations:** 1Department of Orthopaedic Surgery, Okayama Rosai Hospital, Okayama 702-8055, Japan; zygogianniskonstantinos@gmail.com (K.Z.); naveen.sake@gmai.com (N.S.); araoyc@gmail.com (S.A.); fujiwarayoshihiro2004@yahoo.co.jp (Y.F.); taokatakuya@gmail.com (T.T.); thiago.henrique.modesto@gmail.com (T.H.d.M.M.); 2Department of Scoliosis and Spine Department, KAT Hospital, 14-561 Athens, Greece; chatzio69@gmail.com

**Keywords:** C-arm free, minimally invasive spine surgery, adult spinal deformity, lateral access spine surgery, oblique lumbar interbody fusion

## Abstract

*Background and Objectives*: C-arm-free MIS techniques can offer significantly reduced rates of postoperative complications such as inadequate decompression, blood loss, and instrumentation misplacement. Another advantageous long-term aspect is the notably diminished exposure to radiation, which is known to cause malignant changes. This study emphasizes that, in some cases of spinal conditions that require a procedural intervention, C-arm-free MIS techniques hold stronger indications than open surgeries guided by image intensifiers. *Materials and Methods*: This study includes a retrospective analysis and review of various cervical and thoracic spinal procedures, performed in our hospital, applying C-arm-free techniques. The course of this study explains the basic steps of the procedures and demonstrates postoperative and intraoperative results. For anterior cervical surgery, we performed OPLL resection, while for posterior cervical surgery, we performed posterior fossa decompression for Chiari malformation, minimally invasive cervical pedicle screw fixation (MICEPS), and modified Goel technique with C1 lateral mass screw for atlantoaxial subluxation. Regarding the thoracic spine, we performed anterior correction for Lenke type 5 scoliosis and transdiscal screw fixation for diffuse idiopathic skeletal hyperostosis fractures. *Results*: C-arm-free techniques are safe procedures that provide precise and high-quality postoperative results by offering sufficient spine alignment and adequate decompression depending on the case. Navigation can offer significant assistance in the absence of normal anatomical landmarks, yet the surgeon should always appraise the quality of the information received from the software. *Conclusions*: Navigated C-arm-free techniques are safe and precise procedures implemented in the treatment of surgically demanding conditions. They can significantly increase accuracy while decreasing operative time. They represent the advancement in the field of spine surgery and are hailed as the future of the same.

## 1. Introduction

The basic principles of spine treatment can be traced back to ancient Greece, where Hippocrates outlined the importance of spine function. Many centuries afterward, along with medical technology evolution, several attempts were made to increase perioperative accuracy [1]. The first intraoperative guidance pertaining to screw placement consisted of plain radiographs, but that method was abandoned due to low accuracy [2]. In 2013, it was estimated that more than 78% of spine surgeons were successfully using fluoroscopy navigation for intraoperative guidance [3]. Nowadays, C-arm-free navigated MIS techniques offer a variety of advantages, such as minimal soft tissue and muscle detachment, smaller incisions that can reduce pain and infection rates, high precision, and reduced hospitalization [4]. Additionally, as the elderly population is gradually increasing and severe degenerative spine pathology is often associated with opioid overuse and poor conservative results, which overall is a socioeconomic burden to the medical system, the application of those techniques can provide a reliable surgical treatment [5]. On the other hand, a learning curve is expected to be reached in order to obtain an optimal result [6]. The O-arm scan and the navigation registration can be time-consuming and since the equipment and those procedures require familiarization, it can lead to prolonged surgical time. Moreover, the accuracy is not out of the margin of error. If the reference frame is moved, it can significantly compromise the precision, so a second O-arm scan is necessary.

Radiation hazard is also an uprising phenomenon related to fluoroscopy-dependent techniques [7]. Often, prolonged exposure to radiation after years of work can lead to perilous consequences such as malignancy [8]. C-arm-free-navigated techniques can offer zero-level radiation exposure to the operating room staff by performing an O-arm scan. Furthermore, implant misplacement can lead to vessel injury, neurological deterioration, secondary pain, and high revision rates [4]. High-precision fixation is obtained using navigation systems that provide a 3D live image reconstruction during operation [9]. Our study aims to demonstrate the available techniques for C-arm-free-navigated surgery performed in our hospital.

## 2. Occiput–Cervical Spine

### 2.1. Anterior Application

#### OPLL Resection

With the anesthetized patient on the Jackson frame, and the neck in adequate extension. Initially, a classic Smith–Robinson approach is performed to the appropriate cervical levels. After the Caspar pin retractor is placed and distraction is achieved, a special adaptor is used for the placement of the reference frame on the retractor, following an O-arm scan, which offers 3D reconstructed images (Figure 1B). The accuracy can be confirmed by checking at least three different anatomical surfaces. In any doubt, a second scan is strongly recommended. During the next surgical step, discectomies are performed at the superior and inferior disc segments. A navigated high-speed burr is used for bone spur resection in anterior cervical discectomy and fusion (ACDF) or anterior cervical corpectomy and fusion (ACCF). It starts medially to the uncovertebral joints bilaterally in width and to the posterior cortex in-depth (Figure 1C-D). As the anterior approach during decompression is more demanding than the posterior due to the anatomical structures, a navigated pin pointer enables us to confirm the location when needed. The posterior wall of the vertebral body can be easily removed using Kerrison rongeurs or the floating technique. As inadequate decompression can lead to poor postoperative results, the cage position and the level of decompression can be confirmed with a new O-arm scan.

### 2.2. Posterior Application

#### 2.2.1. Posterior Fossa Decompression for Chiari Malformation

The surgical technique starts with an approximate skin incision of 7 cm from the greater occipital protuberance down to the C2 spinous process, followed by a subperiosteal detachment of dorsal cervical muscles from the linea alba, spinous process, and C2 lamina to avoid excessive bleeding. The most convenient point used for reference frame attachment in O-arm scanning is the C2 spinous process (Figure 2A). Next, using a high-speed burr the C1 posterior arch is resected (Figure 2B,C). In some cases, when severe tonsillar herniation exists, we advise performing C2 lamina resection. Excellent knowledge of anatomy is needed to avoid vertebral artery injury. Preoperatively, anatomical variations should also be considered. Thirdly, navigation-assisted craniotomy is performed. Usually, a 3 cm peripheric to foramen magnum is considered adequate. The merit of navigation is the ability to verify the exact bone width and location of the foramen magnum to avoid intraoperative complications. During the last step, duraplasty following ultrasonic monitor verification is performed. In case of inadequate decompression, we recommend the extension of bone resection or duraplasty with the use of artificial dura.

#### 2.2.2. Modified Goel Technique

Stabilization of the C1–C2 level can prove to be challenging due to lower fusion rates compared with the other cervical levels, as the motion in this particular segment is significantly higher. For this approach, we propose C1 LMS with C2 pedicle screw insertion. A careful posterior approach, as described in the midpoint technique, should be made to avoid vessel injury. After that, the ideal entry point should be marked with a high-speed burr without applying strong downward forces at the C2 vertebra. Finally, the trajectory is made using a navigated pedicle probe with appropriate tapping (Figure 3).

#### 2.2.3. Midpoint Technique for C1 Lateral Mass Screw (LMS) Placement

A careful and precise posterior surgical exposure is required for this technique. Following this, retraction of the posterior atlantoaxial venous plexus inferiorly is necessary using a Penfield retractor to separate the posterior arch from the inferior aspect. Firstly, the entry point for this screw is 8 mm anterior from the posterior arch and caudal aspect (midpoint). A navigated high-speed burr with a 2 mm tip is used for precision and safety. Secondly, using a navigated pedicle probe, penetration down to the anterior aspect of the cortex of the C1 anterior arch must be made. At the end, bicortical placement of C1 LMS is performed (Figure 4).

#### 2.2.4. Minimally Invasive Cervical Pedicle Screw Fixation (MICEPS)

During this technique, the patient’s positioning is very important. Thus, a prone position is recommended with a carbon Mayfield cranial support on a Jackson frame to provide additional stability. Initially, a small incision at the prominent C7 spinous process for reference frame attachment is advised, followed by an O-arm scan. Preoperatively, a CT scan verification for pedicle anatomy should be performed and studied. Usually, bilateral 4 cm skin incisions after navigation verification are enough for C3–C6 fixation. At the end, a new O-arm scan should be performed to verify anatomical pedicle screw placement. During open posterior cervical screw fixation, complications such as high infection rates, postoperative pain, and kyphotic deformity are frequently met. Thus, with this lateral minimal technique, it is feasible to reduce the complication rate due to minimal surgical exposure (Figure 5).

## 3. Thoracic Spine

### 3.1. Anterior Application

#### Anterior Correction for Lenke Type 5

A lateral right decubitus position is appropriate for this approach. A left oblique skin incision of approximately 20 cm is made along the 10th or 11th rib. After superficial dissection through abdominal fat, external oblique, internal oblique, and transverse abdominal muscles are split in the mentioned order. Then, a circumferential dissection of the diaphragm is necessary for further exposure. The use of a hand-held retractor minimizes the risk of ureter and vascular injury. Usually, the reference frame’s position, which is attached to the spinous process, must be divided equally between the two surgical edges to maximize accuracy. During the next step, careful screw insertion must be made using navigation for positional accuracy and length safety starting from the furthest fused segment. After the entry point is decided and made using a navigated high-speed burr or an awl, a navigated probe is inserted down to the opposite cortex. For correct measurement of the screw’s length, the opposite cortex must be penetrated. Two of the screws should be placed at a 20-degree angle to provide extra stability for correction maneuvers. Before correction maneuvers are performed, discectomies must be performed to facilitate further release and to achieve adequate fusion. Most of the desired correction can be obtained during the first step by placing the anterior rod and rotating it 90 degrees to create lordosis. Following this, the only way to achieve further correction is with in situ benders. At the end of the correction, an X-ray verification should always be made (Figure 6).

### 3.2. Posterior Application

#### Transdiscal Screw for DISH (Diffuse Idiopathic Skeletal Hyperostosis) Fracture

The patient can be positioned in a prone or lateral decubitus position, though sometimes, the prone position can widen the fracture window, especially in DISH cases. A 1.5 cm incision in the middle line above the most cranial fused spinous process level is recommended for reference frame placement, followed by an O-arm scan. After marking the appropriate pedicle entry point, the probe for the transdiscal screw trajectory should be aimed through the pedicle to the upper endplate and the anterior 1/3 of the inferior endplate of the upper vertebra without reaching the anterior vertebral body wall. The desired angulation should be no more than 25–30 degrees; otherwise, the rod cannot be held firmly well in the tulip. Postoperatively, X-rays should always be taken for screw placement and alignment verification (Figure 7).

## 4. Discussion

The purpose of each surgical procedure, depending on the case, is to achieve solid bone fusion, adequate decompression, and restoration of spinal alignment. Regarding anterior cervical decompression, for ossification of the posterior longitudinal ligament (OPLL), inadequate decompression can lead to poor clinical outcomes and future neurologic deterioration [10]. Using the C-arm-free technique, a surgeon is always able to verify the amount of decompression and the anatomical area [11]. Also, complications such as CSF leakage, which are frequently met, can be reduced. Thus, the merit of this technique is based on the fact that the decompression is far more precise, as the surgeon is able to receive live feedback during the surgery for the location and the amount of decompression [11]. Recent studies suggest that anterior decompression has excellent long-term outcomes for more than 10 years [12]. Onari et al. completed a retrospective study of 30 patients and reported that in 26 of them, OPLL progression was observed after surgery, but due to adequate decompression, there was no clinical deterioration [13].

Likewise, a successful posterior fossa decompression for Chiari malformation type 1 (CM1) is mainly dependent on the amount of bone resection, which sometimes differs from surgeon to surgeon. Anatomical variations such as hypoplasia of basioccipital bone, fused segments, and platybasia can coexist [14,15,16]. While operating under these conditions, a limited craniotomy may result in insufficient decompression, but a large one may lead to the descendance of the cerebellum through the decompressed area. Consequently, with the C-arm-free technique, a surgeon can approach carefully and achieve accurate decompression [17]. In a retrospective study by Limonadi et al., the average surgical time was 249 min [18]; however, with a navigation system, the surgical time can be decreased [17]. 

For atlantoaxial anterior subluxation, achieving stability and bony fusion with minimal soft tissue detachment is of the utmost importance as the patient can be mobilized immediately following surgery without a collar bracing [19]. Although our C-arm-free technique for this condition is also openly approached, the advantage of this method is that surgical exposure is minimal because the lateral mass is not necessarily directly exposed [20]. Usually, the venous plexus is exposed and retracted caudally, and that action results in blood loss. Guo et al. reported that the average blood loss during C1–C2 fixation was 219.1 ± 195.6 mL [21], while with our technique, we reported that the average blood loss was 100 mL [20]. Additionally, the merit of this technique resides in precise entry points, screw placement, and nominal postoperative pain. Another C-arm-free technique for posterior cervical fixation is the minimally invasive cervical pedicle screw fixation (MICEPS) technique. Usually, during pedicle screw placement, the danger lies in a vertebral artery injury [22]. Thus, with this technique, we can safely approach through the pedicle as there is live feedback regarding the trajectory. In a retrospective study, Ishikawa et al. reported that the pedicle breach rate with O-arm navigation was 2.8% [23], which is lower compared with Yukawa et al.’s study, which reported that 3.9% of 620 C-arm-placed pedicle screws demonstrated pedicle breach and 9.2% screw exposure [24].

Thoracic deformity correction for Lenke 5 scoliosis in pediatric patients can prove to be challenging. Scoliosis is a three-dimensional deformity, and adequate correction usually requires bone fusion of mobile segments. Especially in young patients, the preservation of mobile segments results in less back pain and decreased rates of degeneration [25]. In a retrospective study with an 8-year follow up, Wang et al. reported that both posterior and anterior fixation can achieve sufficient correction, but the group with anterior correction had significantly fewer fused segments: 5.1  ±  0.6 compared with the posterior correction group, who had 7.0  ±  1.3 [26]. Moreover, Hirase et al. reported that posterior fixation has higher rates of proximal junctional kyphosis than anterior fixation [27]. Another important issue in thoracic spine surgery is severe osteoporotic patients with unstable fracture patterns. Transdiscal screw fixation can significantly increase the pull-out screw force resulting in more stable fixation. The latest studies reported that bicortical screws can be 1.6–1.8 times stronger than pedicle screw insertion [28]. Moreover, sometimes posterior fixation alone is not sufficient, especially in high-energy burst fractures with kyphotic deformity of the thoracic spine, where anterior column support is mandatory [29]. 

Prolonged exposure to ionizing radiation during complex trauma or deformity cases can gradually lead to indirect DNA lesions and cellular irregularity [30]. During MIS techniques, a surgeon can reduce the soft tissue detachment and decrease the complication rates; however, this does not change the fact that many of them are C-arm dependent. Using C-arm-free techniques, medical staff can perform high-precision surgery and reduce the radiation hazard to a minimum [20]. On the other hand, for multilevel screw fixation, sometimes, more than one O-arm scan may be required. Recent studies suggest that a single O-arm scan can distribute 9 mGy of radiation to a patient, which is equal to 35 s of fluoroscopy [31]. Usually, for a single screw insertion under fluoroscopy, the median time is 10–20 s [32]. Thus, the overall exposure to a patient sometimes may be the same. Furthermore, a perioperative screen image is established and fixed once the O-arm scan is made. After that, a surgeon is not able to visualize any dynamic changes made during the operation, and sometimes, we still have to rely on X-rays [33]. Surgeons should always be aware that accuracy depends on the stability of the reference frame and the liability of the system. If the reference frame is moved or there is any doubt of software malfunction, another scan should be performed or abandoned (Table 1).

## 5. Conclusions

The minimally invasive C-arm-free technique provides the benefit of precision with zero radiation exposure. In expert hands, it can significantly reduce surgical time and blood loss. In the near future, O-arm-navigated spine surgery will prove to be a golden tool for surgeons who perform a large number of operations annually.

## Figures and Tables

**Figure 1 medicina-59-01779-f001:**
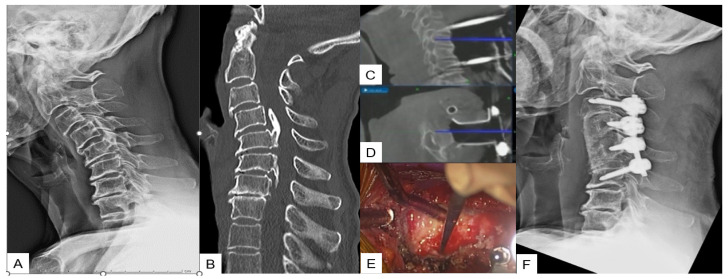
Case 1: 82-year-old male; cervical OPLL, anterior cervical corpectomy and fusion. (**A**) Preoperative cervical lateral radiogram. (**B**) Preoperative CT sagittal reconstruction image. (**C**) Intraoperative sagittal navigation image. (**D**) Intraoperative axial navigation image. (**E**) Intraoperative image. (**F**) Postoperative cervical lateral radiogram.

**Figure 2 medicina-59-01779-f002:**
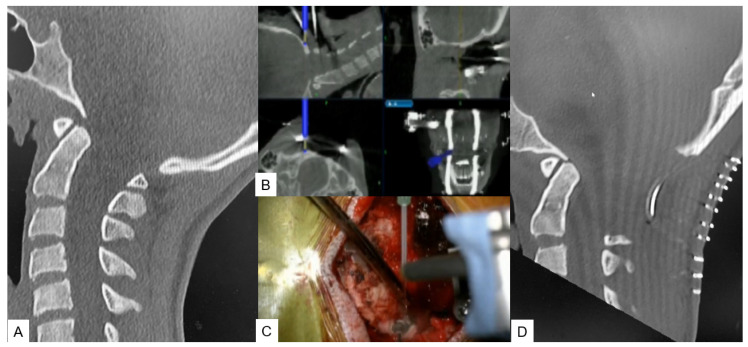
Case 2: 14-year-old female; Chiari malformation, foramen magnum decompression. (**A**) Preoperative occipitocervical CT sagittal reconstruction. (**B**) Intraoperative navigation image. (**C**) Intraoperative image. (**D**) Postoperative occipitocervical CT sagittal reconstruction.

**Figure 3 medicina-59-01779-f003:**
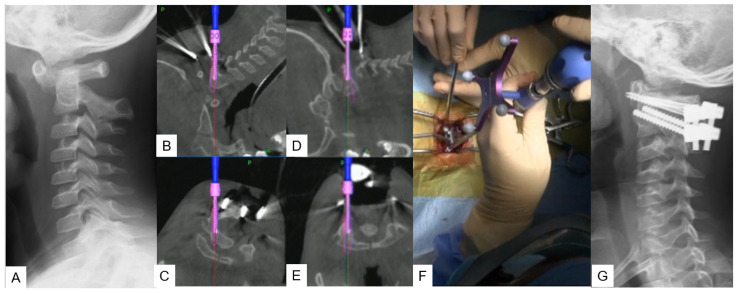
Case 3: 5-year-old male; Down syndrome, anterior atlantoaxial subluxation, modified Goel technique. (**A**) Preoperative lateral cervical radiogram. (**B**,**C**) Intraoperative navigation image of C2 pedicle screw insertion. (**D**,**E**) Intraoperative navigation image of C1 lateral mass screw insertion. (**F**) Intraoperative image. (**G**) Postoperative lateral cervical radiogram.

**Figure 4 medicina-59-01779-f004:**
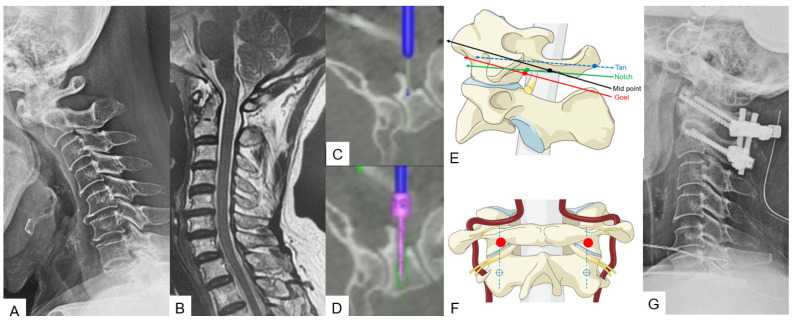
Case 4: 68-year-old male; anterior atlantoaxial subluxation, modified Goel fixation. (**A**) Preoperative cervical lateral radiogram. (**B**) Preoperative midsagittal T2 weighted MR imaging. (**C**) Intraoperative navigated high-speed burr. (**D**) Intraoperative lateral mass screwing. (**E**) Various C1 lateral mass entry points in lateral. (**F**) Insertion point of mid-point technique (red circle). (**G**) Postoperative cervical lateral radiogram.

**Figure 5 medicina-59-01779-f005:**
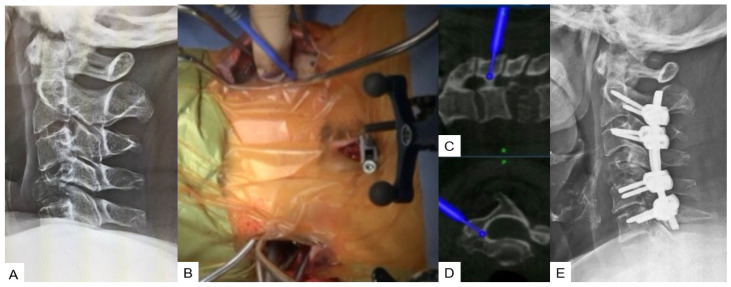
Case 5: 62-year-old female; Brest cancer, C4 metastasis, C2–C6 posterior fusion. (**A**) Preoperative cervical lateral radiogram. (**B**) Intraoperative image. (**C**) Intraoperative sagittal navigation image. (**D**) Intraoperative axial navigation image. (**E**) Postoperative cervical lateral radiogram.

**Figure 6 medicina-59-01779-f006:**
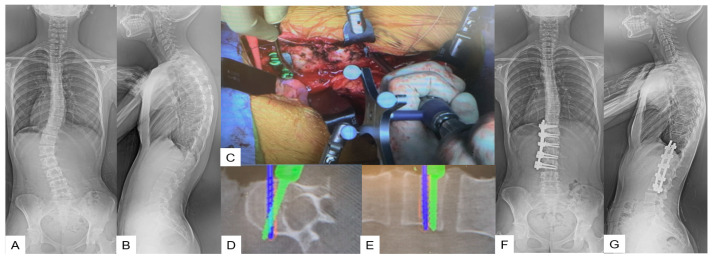
Case 6: 15-year-old female; adolescent idiopathic scoliosis LT11-L3enke Type 5C, T12-L3 anterior fusion. Preoperative 46 degrees of Cobb angle (T12-L3) became 1 degree. (**A**) Preoperative posteroanterior spine radiogram. (**B**) Preoperative lateral spine radiogram. (**C**) Intraoperative image. (**D**) Intraoperative axial navigation image. (**E**) Intraoperative coronal navigation image, (**F**) Postoperative posteroanterior spine radiogram. (**G**) Postoperative lateral spine radiogram.

**Figure 7 medicina-59-01779-f007:**
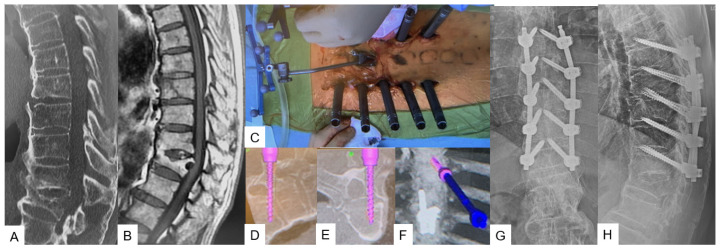
Case 8: 85-year-old male; T8 fracture of diffuse idiopathic skeletal hyperostosis (DISH) patient. (**A**) Preoperative sagittal reconstruction CT. (**B**) Preoperative sagittal T1 weighted MR imaging. (**C**) Intraoperative image. (**D**) Intraoperative sagittal navigation image. (**E**) Intraoperative axial navigation image. (**F**) Intraoperative 3D navigation image. (**G**) Postoperative anteroposterior radiogram. (**H**) Postoperative lateral radiogram.

**Table 1 medicina-59-01779-t001:** Advantages and disadvantages of the C-arm-free technique.

Advantage	Disadvantage
Zero exposure of OR staff to radiation	Absence of dynamic change in a perioperative image
Excellent accuracy	Steep learning curve
Less surgical time	System malfunction

## Data Availability

The data presented in this study are available in the article.

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
