# Peer review of "A C-Arm-Free Minimally Invasive Technique for Spinal Surgery: Cervical and Thoracic Spine"

_medicina, 2023, doi:10.3390/medicina59101779_

Round 1

Reviewer 1 Report

The articleby Tanaka et al. "C-arm Free Minimally Invasive Technique for Spinal Surgery: Cervical and Thoracic Spine" covers a potentially interesting and emerging topic related to the complex spine surgery. In this sense, this remains to be potentially interesting for the medicina readers. I regard the main point of this paper as highly attractive as well as the results are clearly presented. The text does not contain any major errors, therefore I
have some minor comments and recommendations:

1. There is a need to provide slightly more expanded introduction shortly
mentioning/describing pathogenesis of degenerative spine diseasesand its impact of modern healthcare/pharmacoeconomics.
2. The figure summarizing and clarifying the results should be added.
3. Following references should be added and properly cited within the main text:

- Kubaszewski Ł, Wojdasiewicz P, Rożek M, Słowińska IE, Romanowska-Próchnicka K, Słowiński R, Poniatowski ŁA, Gasik R. Syndromes with chronic non-bacterial osteomyelitis in the spine. Reumatologia. 2015;53(6):328-36. doi: 10.5114/reum.2015.57639.

-Momin AA, Steinmetz MP. Evolution of Minimally Invasive Lumbar Spine Surgery. World Neurosurg. 2020 Aug;140:622-626. doi: 10.1016/j.wneu.2020.05.071. - Turczyn P, Wojdasiewicz P, Poniatowski ŁA, Purrahman D, Maślińska M, Żurek G, Romanowska-Próchnicka K, Żuk B, Kwiatkowska B, Piechowski-Jóźwiak B, Szukiewicz D. Omega-3 fatty acids in the treatment of spinal cord injury: untapped potential for therapeutic intervention? Mol Biol Rep. 2022 Nov;49(11):10797-10809. doi: 10.1007/s11033-022-07762-x. -Patel PD, Canseco JA, Houlihan N, Gabay A, Grasso G, Vaccaro AR. Overview of Minimally Invasive Spine Surgery. World Neurosurg. 2020 Oct;142:43-56. doi: 10.1016/j.wneu.2020.06.043.   4. In some places the use of English could be improved on.

Completing this gaps will have an impact on the understanding the aim of the study and, from my point of view, is absolutely necessary.

In some places the use of English could be improved on.

Author Response

We appreciate your valuable efforts.

  1. There is a need to provide slightly more expanded introduction shortly
    mentioning/describing pathogenesis of degenerative spine diseases and its impact of modern healthcare/pharmacoeconomics.

Thank you for your productive comments , according to your advice we changed the introduction as followed:

The basic principles of spine treatment can be traced back in ancient Greece where Hippocrates outlined the importance of spine function. Many centuries afterwards along with the medical technology evolution, several attempts were made to increase the perioperative accuracy [1]. The first intraoperative guidance pertaining to screw placement consisted of plain radiographs but that method was abandoned due to low accuracy [2]. In 2013, it was estimated that more than 78% of spine surgeons were successfully using fluoroscopy navigation for intraoperative guidance [3]. Nowadays, C-arm free navigated MIS techniques offer a variety of advantages, such as, minimal soft tissue and muscle detachment, smaller incisions that can reduce pain and infection rates, high precision, and reduced hospitalization [4]. Additionally, as the elderly population is gradually increasing and severe degenerative spine pathology is often associated with opioid overuse and poor conservative results, which overall is a socioeconomic burden to the medical system, the application of those techniques can provide a reliable surgical treatment [5]. On the other hand, a learning curve is expected to be reached in order to obtain an optimal result [6]. The O-arm scan and the registration of navigation can be time consuming and since the equipment and those procedures require familiarization, it can lead to prolonged surgical time. Moreover, the accuracy is not out of the margin of error. If the reference frame is moved, it can significantly compromise the precision, so a second O-arm scan is necessary.

  1. The figure summarizing and clarifying the results should be added.

We appreciate your advice, the table 1 was added.

Table 1 Advantages and disadvantages of C-arm-free technique

Advantage

Disadvantage

Zero exposure to radiation to OR staff

Absence of dynamic change of perioperative image

Excellent accuracy

Steep learning curve

Less surgical time

System malfunction

  1. Following references should be added and properly cited within the main text:

Thank you for your recommendation, reference 2 and 4 were added as advised.

  1. In some places the use of English could be improved on.

Thank you, according to your advice, we tried to improve the English language.

Reviewer 2 Report

I would like to commend the authors for their comprehensive and insightful work on this topic.

Abstract and manuscript

1. The abstract states "C-arm free MIS techniques can offer significantly reduced rates of postoperative complications such as blood loss and hospitalization time" but later only mentions "Blood loss, postoperative pain and wound infection" as aspects that must be considered. It's unclear if the C-arm free MIS techniques directly impact these aspects based on this abstract.

2. The introduction sets up a clear distinction between the benefits of MIS and the drawbacks related to radiation exposure from fluoroscopy-dependent techniques. However, there's no direct comparison provided about whether C-arm free navigated MIS techniques specifically counter these radiation-related concerns

3. advantages of C-arm free navigated MIS techniques are listed, potential disadvantages or challenges are not discussed, giving a one-sided view in the introduction

4. Implant misplacement consequencesare mentioned but there's no direct link drawn between this and the use of C-arm free navigated MIS techniques. Does the MIS prevent or reduce the risk of implant misplacement?

5. in the discussion section, the authors does not fully integrate the comparative outcomes between traditional and C-arm free techniques. While they mention specific benefits of the C-arm free technique, the direct comparison between the two in terms of outcomes, pros, and cons is missing.

6. The discussion lacks a clear synthesis or summary of the main findings

Surgical techniques and figures are adequate

Conclusions are ok

References are ok

English

1. "The course of the study explains the basic steps of the procedures and demonstrating postoperative or intra-operative results"-> It should be "and demonstrates" instead of "and demonstrating."

2. "They represent the advancement in the field spine surgery" - It's missing the word "of"

3. "First introduced by Long and Matthews in 1995, minimally invasive spine surgery (MIS) nowadays is becoming more and more popular [1]." It should be revised to either "nowadays has become" or just "is becoming."

4. "Our study aims to feature the available techniques of MIS and demonstrate postoperative examples for each category.". The use of the word "feature" is ambiguous.

5. "Onari et al in a retrospective study..." - Missing comma: "Onari et al, in a retrospective study..."

6. "Yukawa’s et al study who reported..." It should be "Yukawa et al.'s study, which reported..."

7. Consistency in writing "C-arm" is not maintained; sometimes it is written as "c-arm", and other times it's "C-arm"

Author Response

We appreciate your valuable efforts.

  1. The abstract states "C-arm free MIS techniques can offer significantly reduced rates of postoperative complications such as blood loss and hospitalization time" but later only mentions "Blood loss, postoperative pain and wound infection" as aspects that must be considered. It's unclear if the C-arm free MIS techniques directly impact these aspects based on this abstract.

We appreciate your effort. Abstract was change according to your advice

Abstract: Background and Objectives: C-arm free MIS techniques can offer significantly reduced rates of postoperative complications such as inadequate decompression, blood loss and instrumentation misplacement. Another advantageous long-term aspect is the notably diminished exposure to radiation which is known to cause malignant changes. This study emphasizes that, in some cases of spinal conditions which require a procedural intervention, C-arm free MIS techniques hold stronger indications than open surgeries guided by image intensifiers.

  1. The introduction sets up a clear distinction between the benefits of MIS and the drawbacks related to radiation exposure from fluoroscopy-dependent techniques. However, there's no direct comparison provided about whether C-arm free navigated MIS techniques specifically counter these radiation-related concerns

Please accept our apology to your request because this is a review paper of our techniques. We did not perform any comparative study. Your concern is integrated in the discussion part.

  1. advantages of C-arm free navigated MIS techniques are listed, potential disadvantages or challenges are not discussed, giving a one-sided view in the introduction

We appreciate your advice. According to your request we added a summary table and a text in the discussion part.

On the other hand, for multilevel screw fixation sometimes more than one O-arm scan may be required. Recent studies suggest that a single O-arm scan can distribute 9 mGy of radiation to the patients which is equal to 35 seconds of fluoroscopy [32]. Usually for a single screw insertion under fluoroscopy the median time is 10-20 seconds [33]. Thus the overall exposure to the patient sometimes may be the same. Furthermore, the perioperative screen image is established and fixed once the O-arm scan is made. After that the surgeon is not able to visualize any dynamic changes made during the operation and sometimes we still have to rely on x-rays [34]. The surgeon should always be aware that accuracy is on the stability of the reference frame and the system`s liability. If the reference frame is moved or there is any doubt of software malfunction another scan should be performed or abandoned.

  1. Implant misplacement consequencesare mentioned but there's no direct link drawn between this and the use of C-arm free navigated MIS techniques. Does the MIS prevent or reduce the risk of implant misplacement?

We apologize for the inconvenience. Navigation reduces the risk of misplacement and not the MIS procedure.

  1. in the discussion section, the authors does not fully integrate the comparative outcomes between traditional and C-arm free techniques. While they mention specific benefits of the C-arm free technique, the direct comparison between the two in terms of outcomes, pros, and cons is missing.

According to your advice we added Table 1.

Table 1 Advantages and disadvantages of C-arm-free technique

Advantage

Disadvantage

Zero exposure to radiation to OR staff

Absence of dynamic change of perioperative image

Excellent accuracy

Steep learning curve

Less surgical time

System malfunction

  1. The discussion lacks a clear synthesis or summary of the main findings

We understand your concern and appreciate it. But this paper is a demonstration of surgical techniques with less radiation for OR staff and surgeons.

Comments on the Quality of English Language

English

  1. "The course of the study explains the basic steps of the procedures and demonstrating postoperative or intra-operative results"-> It should be "and demonstrates" instead of "and demonstrating."
  2. "They represent the advancement in the field spine surgery" - It's missing the word "of"
  3. "First introduced by Long and Matthews in 1995, minimally invasive spine surgery (MIS) nowadays is becoming more and more popular [1]." It should be revised to either "nowadays has become" or just "is becoming."
  4. "Our study aims to feature the available techniques of MIS and demonstrate postoperative examples for each category.". The use of the word "feature" is ambiguous.
  5. "Onari et al in a retrospective study..." - Missing comma: "Onari et al, in a retrospective study..."
  6. "Yukawa’s et al study who reported..." It should be "Yukawa et al.'s study, which reported..."
  7. Consistency in writing "C-arm" is not maintained; sometimes it is written as "c-arm", and other times it's "C-arm"

We corrected according to your advice.

Round 2

Reviewer 2 Report

I agree with the authors answers and changes made. I accept their work as it is now.